# Cryo-EM analysis of PIP$_2$ regulation in mammalian GIRK channels

Yiming Niu[†], Xiao Tao[†], Kouki K Touhara, Roderick MacKinnon*

Laboratory of Molecular Neurobiology and Biophysics, The Rockefeller University, Howard Hughes Medical Institute, New York, United States

**Abstract** G-protein-gated inward rectifier potassium (GIRK) channels are regulated by G proteins and PIP$_2$. Here, using cryo-EM single particle analysis we describe the equilibrium ensemble of structures of neuronal GIRK2 as a function of the C8-PIP$_2$ concentration. We find that PIP$_2$ shifts the equilibrium between two distinguishable structures of neuronal GIRK (GIRK2), extended and docked, towards the docked form. In the docked form the cytoplasmic domain, to which G$_{βγ}$ binds, becomes accessible to the cytoplasmic membrane surface where G$_{βγ}$ resides. Furthermore, PIP$_2$ binding reshapes the G$_{βγ}$ binding surface on the cytoplasmic domain, preparing it to receive G$_{βγ}$. We find that cardiac GIRK (GIRK1/4) can also exist in both extended and docked conformations. These findings lead us to conclude that PIP$_2$ influences GIRK channels in a structurally similar manner to Kir2.2 channels. In Kir2.2 channels, the PIP$_2$-induced conformational changes open the pore. In GIRK channels, they prepare the channel for activation by G$_{βγ}$.

## Introduction

The inward rectifier K$^+$ (Kir) channels were originally named for their rectifying current-voltage relationship (*Hagiwara et al., 1976*; *Hagiwara and Takahashi, 1974*; *Hodgkin and Horowicz, 1959*; *Noble, 1965*). Today, this class of ion channels is defined by characteristic structural features encoded by the Kir family of related genes (*Hibino et al., 2010*). All Kir channels are tetramers of identical or related subunits that encode a K$^+$ selectivity filter-containing transmembrane pore (TMD for transmembrane domain) and a cytoplasmic domain (CTD) (*Tao et al., 2009*; *Whorton and MacKinnon, 2011*). The TMD and CTD are covalently linked by a tether, called here the TMD-CTD linker (*Tao et al., 2009*; *Whorton and MacKinnon, 2011*). In eukaryotic cells, Kir channels underlie many physiological processes, including neuronal electrical activity, electrolyte homeostasis in the kidney, insulin secretion, and heart rate control (*Hibino et al., 2010*).

To fulfill their many biological roles, different eukaryotic Kir channels respond to unique ligands. However, as far as we know, they all respond to the signaling lipid phosphatidylinositol 4,5-bisphosphate (PIP$_2$) (*Hibino et al., 2010*; *Hilgemann et al., 2001*; *Huang et al., 1998*; *Stanfield et al., 2002*). In fact, the two features all eukaryotic Kir channels have in common are K$^+$ selectivity and responsiveness to PIP$_2$. A specific mechanism for PIP$_2$ regulation of Kir2.2 has been proposed (*Hansen et al., 2011*; *Tao et al., 2009*). As depicted (*Figure 1A*), in the absence of PIP$_2$ the CTD is disengaged from the TMD, resulting in an 'extended' conformation (*Tao et al., 2009*). Upon PIP$_2$ binding, the TMD-CTD loop forms a helix and the CTD engages the TMD to form a 'docked' conformation in which the channel opens (*Hansen et al., 2011*).

G-protein-gated inward rectifier K$^+$ (GIRK) channels are activated by both PIP$_2$ and G$_{βγ}$ (*Huang et al., 1998*; *Logothetis et al., 1987*; *Logothetis and Zhang, 1999*; *Sui et al., 1998*). Detailed mechanistic studies in a reconstituted system using a neuronal isoform of GIRK (GIRK2) showed that both PIP$_2$ and G$_{βγ}$ are required to open the pore (*Wang et al., 2014*). Crystal structures of GIRK2 in the absence and presence of PIP$_2$ (and in the presence of G$_{βγ}$) did not show a change in the relationship between the CTD and TMD as was seen in Kir2.2 (*Hansen et al., 2011*; *Tao et al.,*

*For correspondence:
mackinn@mail.rockefeller.edu

[†]These authors contributed equally to this work

Competing interests: The authors declare that no competing interests exist.

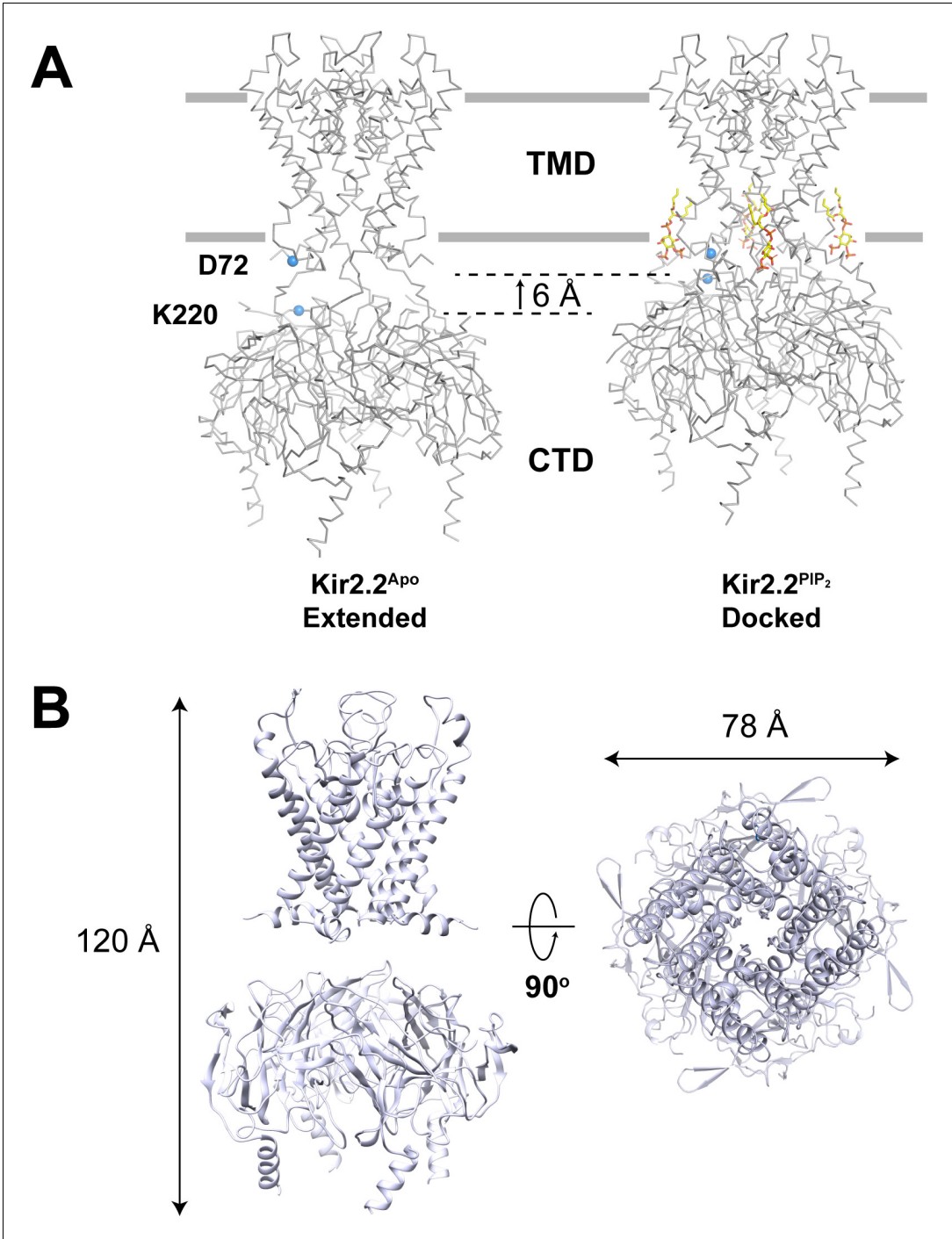

**Figure 1.** GIRK2 adopts an extended conformation in the absence of $PIP_2$. (**A**) Conformational changes upon $PIP_2$ binding in the Kir2.2 channel viewed from side with the extracellular side above (Left: the extended conformation without $PIP_2$, PDB: 3JYC. Right: the docked conformation upon $PIP_2$ binding, PDB: 3SPI). Four $PIP_2$ molecules are shown as sticks and colored according to atom type: carbon, yellow; phosphorous, orange; and oxygen, red. The CTD translates towards the TMD by 6 Å upon $PIP_2$ binding. A set of reference atoms (Asp72 and Lys220 α-carbons) are highlighted as blue spheres in each structure. (**B**) Side and top views of the cryo-EM structure of the GIRK2 channel in an extended conformation.

The online version of this article includes the following figure supplement(s) for figure 1:

**Figure supplement 1.** Cryo-EM analysis of the GIRK2 channel in the extended conformation, related to *Figure 1*.

*Figure 1 continued*

**Figure supplement 2.** Cryo-EM densities for selected regions of the GIRK2 extended conformation (contour level 6.5 in Coot), related to *Figure 1*.

**Figure supplement 3.** Structural comparison of the apo GIRK2 determined by cryo-EM (gray) and X-ray crystallography (salmon, PDB: 3SYO), related to *Figure 1*.

---

*2009*; *Whorton and MacKinnon, 2011*; *Whorton and MacKinnon, 2013*). In this manuscript, we study the structural effects of $PIP_2$ on neuronal GIRK2 and cardiac GIRK1/4 using cryo-electron microscopy (cryo-EM) and correlate these effects with known properties of $PIP_2$ activation.

## Results

### GIRK2 adopts an extended conformation in the absence of $PIP_2$

*Figure 1B* shows the structure of the mouse GIRK2 channel in the absence of $PIP_2$ (CryoEM$^{Apo}$) at a resolution of 3.9 Å by cryo-EM single particle analysis. The corresponding cryo-EM map is shown in *Figure 1—figure supplements 1* and *2*. In the absence of $PIP_2$ it is clear that GIRK2 can adopt a conformation in which the CTD is disengaged from the TMD and the TMD-CTD linker has to be extended to account for the separation between the TMD and CTD. Thus, in the absence of $PIP_2$, the global conformation of GIRK2 appears much like the Kir2.2 channel in the absence of $PIP_2$ (i.e. both channels adopt an extended conformation) (*Figure 1A and B*; *Tao et al., 2009*).

Using X-ray crystallography, we had determined a structure of GIRK2 in the absence of $PIP_2$ (Crystal$^{Apo}$) that did not adopt this extended conformation (*Figure 1—figure supplement 3A*; *Whorton and MacKinnon, 2011*). Detailed differences in the structures of CryoEM$^{Apo}$ and Crystal$^{Apo}$ are shown (*Figure 1—figure supplement 3B and C*). The crystal lattice offers a possible explanation for conformational differences between CryoEM$^{Apo}$ and Crystal$^{Apo}$ (*Figure 1—figure supplement 3D*). Symmetry-related tetramers in the crystal contact each other through ~500 Å$^2$ buried surface area on the CTD (PISA server) (*Krissinel and Henrick, 2007*), as shown in the inset of *Figure 1—figure supplement 3D*. Because in the extended conformation the CTD and TMD are free to move with respect to each other without hindrance, these two structured regions may have been pushed together when the crystal was formed. Single particle cryo-EM, without the potential interference of lattice contacts, might provide a structure that more accurately portrays a GIRK2 channel in the cell membrane.

GIRK2 and Kir2.2 are structurally similar channels, but differ functionally in an important aspect. $PIP_2$ is necessary and sufficient to open Kir2.2, but must operate in conjunction with $G_{\beta\gamma}$ to open GIRK2. This important distinction notwithstanding, the difference between Crystal$^{Apo}$ and CryoEM$^{Apo}$ conformations leads us to ask which conformation better reflects reality in the membrane? To pursue this question further, we used cryo-EM single particle analysis to study the dependence of the GIRK2 conformation as a function of $PIP_2$ concentration.

### GIRK2 conformation as a function of $PIP_2$ concentration

GIRK2 channels were vitrified in the presence of $PIP_2$ (soluble C8-$PIP_2$) concentrations ranging from 0 mM to 1.0 mM. Data were collected and analyzed using the approach applied to the Na$^+$ dependence of the Slo2 K$^+$ channel conformation (*Hite and MacKinnon, 2017*). Images from all concentrations were merged into a single 'titration dataset' and 3D refinement was carried out in RELION (*Scheres, 2012*). Particles were then classified (five classes requested) without refinement of angles or translations. Four classes (1-4) were similar to each other and showed a disengaged CTD (i.e. extended conformation) with an unresolved TMD-CTD linker (*Figure 2A*). Class 5 showed an engaged CTD (i.e. CTD-docked conformation) and a visible TMD-CTD linker (*Figure 2A*). The fraction of channels contributing to the CTD-docked conformation (class 5) increased as $PIP_2$ concentration increased (*Figure 2B and C*).

To further examine the 3D classification result, we performed classification five times independently and compared the outcome. As shown in *Figure 2—figure supplement 1A*, the fraction of channels contributing to the docked class at each $PIP_2$ concentration remained fairly constant for the five independent runs, indicating that the classification algorithm yields a reproducible outcome. In

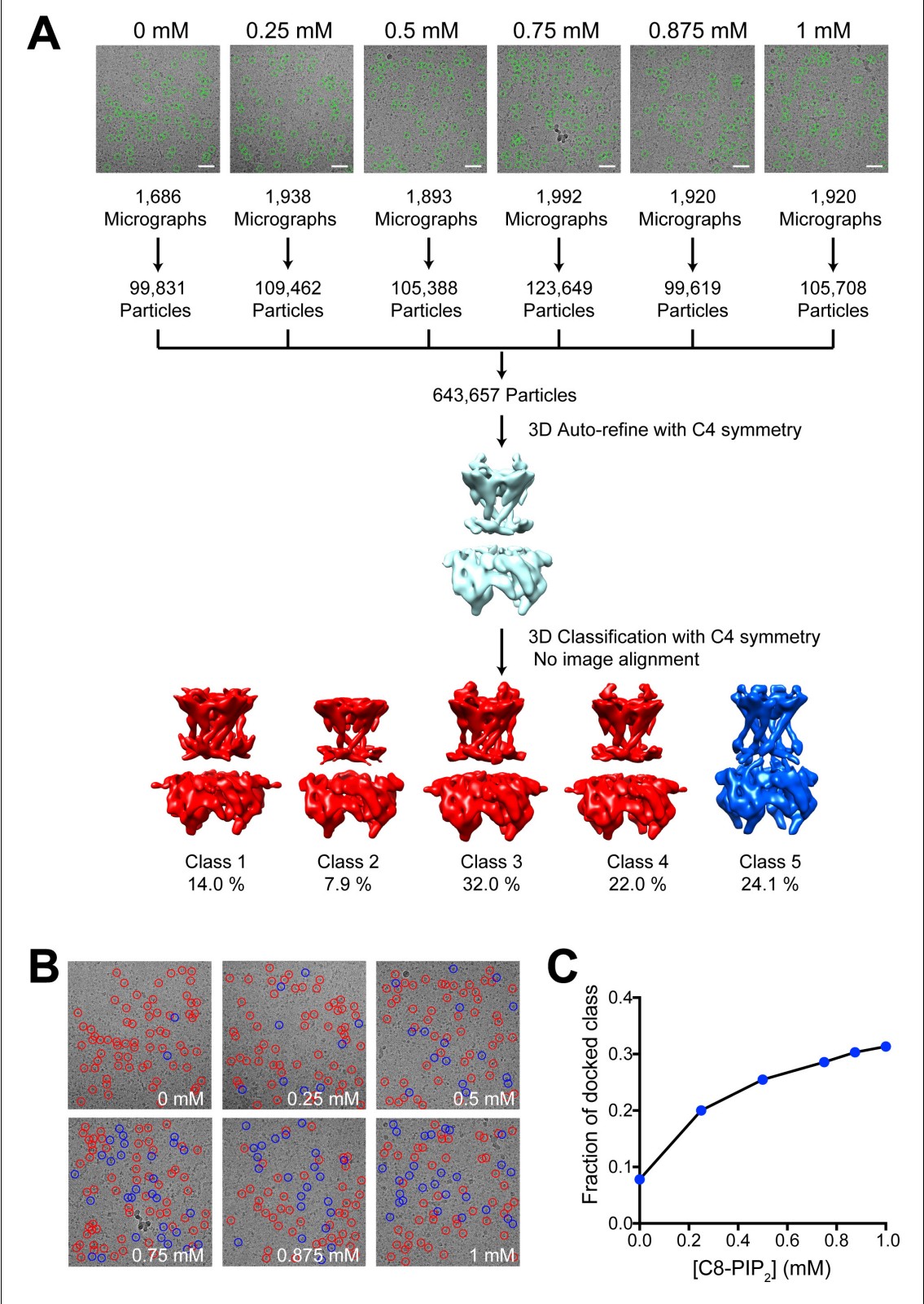

**Figure 2.** GIRK2 conformation as a function of PIP₂ concentration. (**A**) Structural titration image analysis workflow. Representative images of GIRK2 channels recorded in the presence of 0, 0.25, 0.5, 0.75, 0.875 or 1 mM C8-PIP₂. GIRK2 particles were automatically selected from the images (green circles). The extracted particles from the respective images were combined into a single titration dataset for 3D refinement with C4 symmetry in RELION. Using the angles and translations obtained from the 3D refinement, the particles from the titration dataset were classified into five classes. The
*Figure 2 continued on next page*

*Figure 2 continued*

extended classes are colored red and the docked class blue. (B) Representative cryo-EM images of GIRK2 in the presence of 0, 0.25, 0.5, 0.75, 0.875 or 1 mM C8-PIP$_2$. Particles marked with a red circle were classified as extended, and those with a blue circle were classified as docked. (C) The fraction of particles classified as the docked conformation of GIRK2 is plotted against the concentration of C8-PIP$_2$.

The online version of this article includes the following figure supplement(s) for figure 2:

**Figure supplement 1.** Reproducibility of the 3D classification of extended and docked classes of the GIRK2 channel, related to *Figure 2*.

addition, of all the channels classified as docked, more than 80% were classified as such four or five times (*Figure 2—figure supplement 1B*). These data indicate that the population of channels in the docked conformation is positively correlated with the concentration of PIP$_2$.

*Figure 2C* shows that the fraction of docked channels increased from approximately 0.08 to 0.30 when PIP$_2$ is increased from 0 mM to 1.0 mM. We previously showed that the activity of GIRK2 channels in membranes increases as a function of PIP$_2$ concentration with an activation constant ~15 μM (and Hill coefficient ~3.1) (*Wang et al., 2014*). The large difference in PIP$_2$ activity between these studies likely originates in the difference between the detergent micelle and the lipid membrane environment. PIP$_2$ partitions into membranes (differently than into detergent micelles) and therefore the local concentrations of PIP$_2$ are unknown.

It is also notable that a fraction (~0.08) of docked channels exists in the absence of PIP$_2$ (*Figure 2C*). In functional studies in membranes we found that the open fraction of GIRK2 channels in the absence of PIP$_2$ (but in the presence of G$_{\beta\gamma}$) is probably less than 0.01 (*Wang et al., 2014*). Again, there is an uncertainty that precludes direct comparison of these numbers: we do not know what fraction of docked channels are open. In other words, docking may be a necessary but not necessarily a sufficient condition to open the pore.

We found previously that the presence of 32 mM Na$^+$ substantially increases the open probability of GIRK2 at all concentrations of PIP$_2$ examined (*Wang et al., 2014*). The experiments behind *Figure 2* were carried out in the absence of added Na$^+$. It is possible that addition of Na$^+$ would increase the fraction of docked channels.

Presently, we conclude that formation of the docked channel is positively correlated with increased PIP$_2$ concentration.

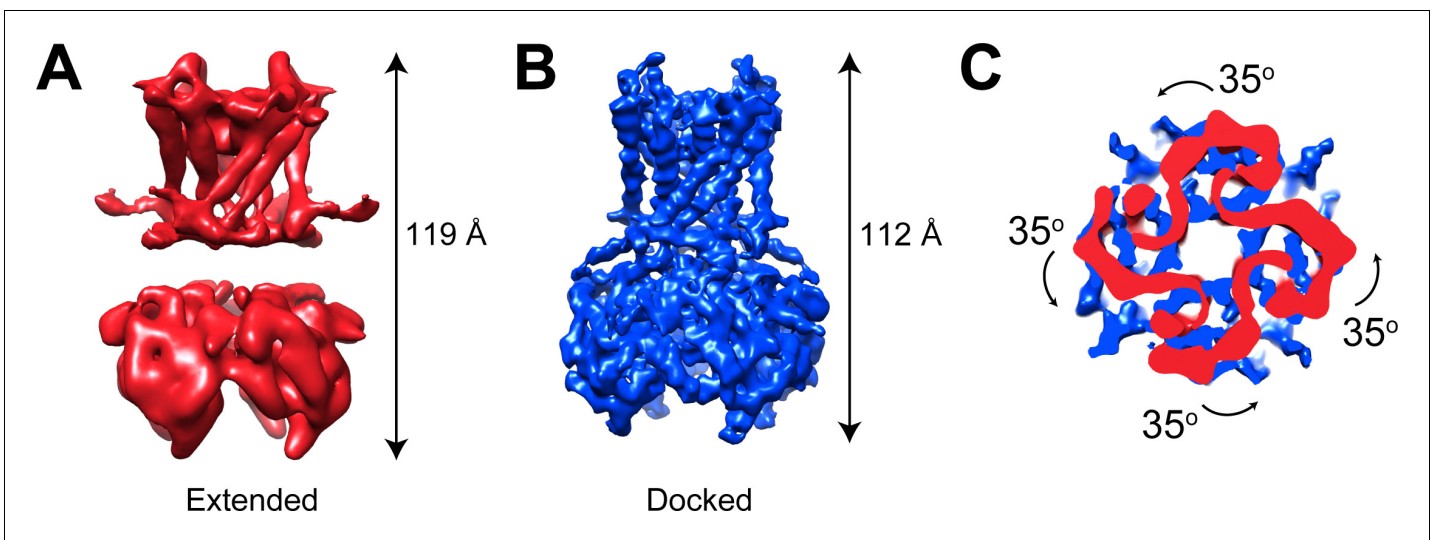

**Figure 3.** GIRK1/4 channels also form extended and docked conformations. (**A and B**) Side views of the cryo-EM density map of the extended (**A**) and docked (**B**) conformations of the GIRK1/4 channel. (**C**) Top view of the CTD regions aligned with respect to the TMD reveals a 35° rotation from extended to docked conformations.

The online version of this article includes the following figure supplement(s) for figure 3:

**Figure supplement 1.** Cryo-EM analysis of the GIRK1/4 channel in the presence of 0.5 mM C8-PIP$_2$, related to *Figure 3*.

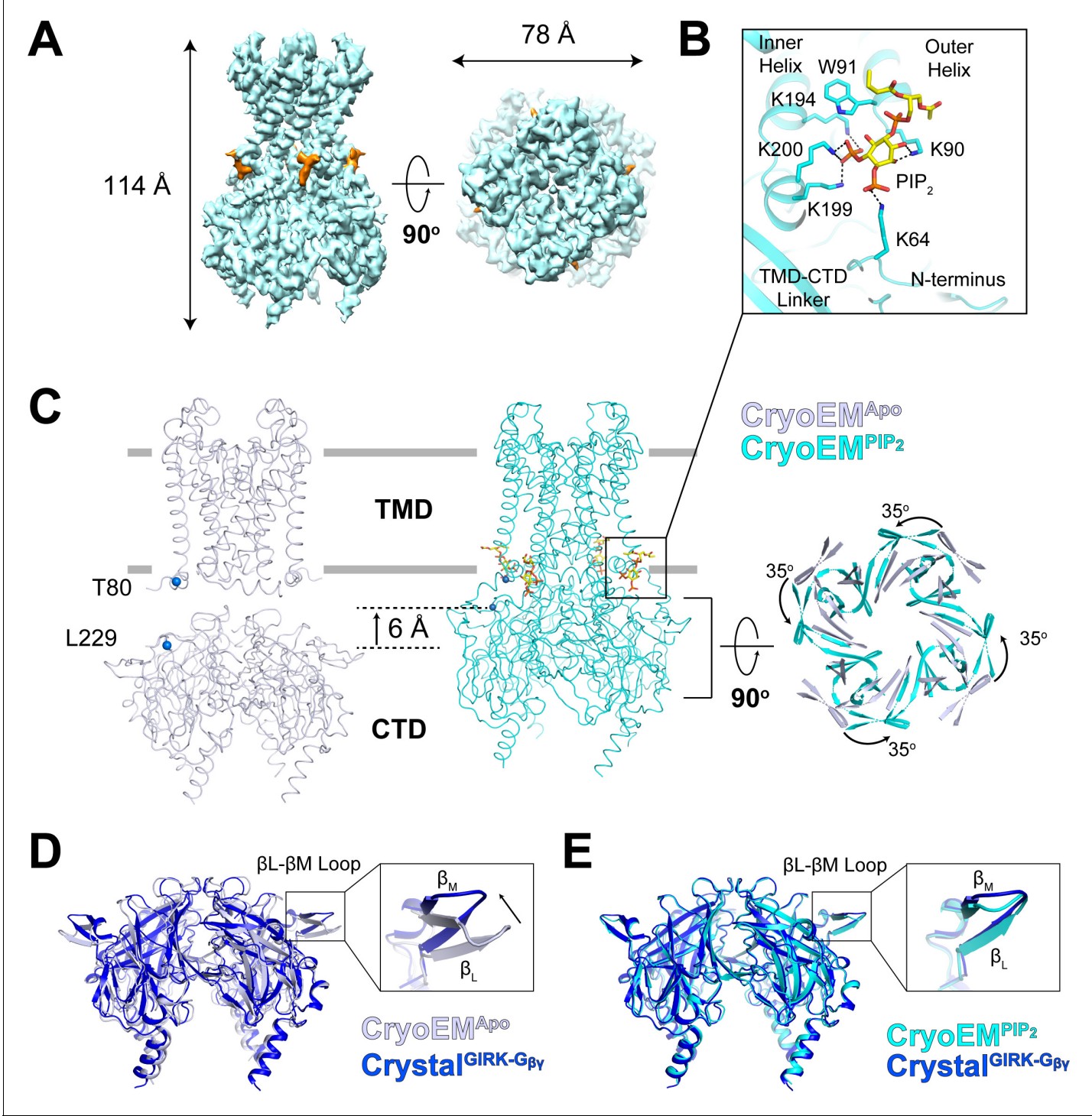

**Figure 4.** Conformational changes between the docked and extended GIRK2 channel upon PIP$_2$ binding. (**A**) Side and top views of the cryo-EM density map of the docked conformation of GIRK2 channel (cyan) with four bound PIP$_2$ molecules (orange). The PIP$_2$ acyl chains were only partially resolved. (**B**) Close-up view of the PIP$_2$ binding pocket. PIP$_2$ is shown as sticks and colored according to atom type (carbon, yellow; phosphorous, orange; and oxygen, red). The PIP$_2$ interacting residues are also shown as sticks (carbon, cyan and nitrogen, blue). (**C**) Comparison of the extended (gray, no PIP$_2$) and the docked (cyan, with 4 PIP$_2$ bound) structures. The channel is viewed from the side with the extracellular side above. The lipid bilayer boundaries are shown as grey bars. Four PIP$_2$ molecules are shown as sticks and colored as in panel (**B**). The PIP$_2$ molecule in a similar orientation as in (**B**) is outlined by a black box. Upon PIP$_2$ binding, the CTD in the extended structure translates towards the TMD by 6 Å accompanied by a 35° rotation viewed from the extracellular side. (**D and E**) Local conformational changes at the binding site for G$_{\beta\gamma}$ in the βL-βM loop of GIRK2 CTD are shown by

*Figure 4 continued on next page*

*Figure 4 continued*

structural superposition. The CTD region of GIRK2 CryoEM$^{Apo}$, CryoEM$^{PIP2}$ and Crystal$^{GIRK-G\beta\gamma}$ structures are colored gray, cyan, and blue, respectively. Rearrangement of the βL-βM loop is indicated by an arrow.

The online version of this article includes the following figure supplement(s) for figure 4:

**Figure supplement 1.** Cryo-EM analysis of the GIRK2 channel in the docked conformation, related to *Figures 2* and *4*.

**Figure supplement 2.** Cryo-EM densities for selected regions of the GIRK2 docked conformation (contour level 8.0 in Coot), related to *Figures 2* and *4*.

**Figure supplement 3.** PIP$_2$ binding pocket, related to *Figure 4*.

## GIRK1/4 channels also form extended and docked conformations

GIRK2 channels predominate in the nervous system while GIRK4 (Kir3.4) and heteromultimeric GIRK1/4 (Kir3.1/Kir3.4) channels function in the cardiovascular system where they regulate heart rate through parasympathetic nervous system control (*Corey and Clapham, 1998*; *Karschin et al., 1996*; *Krapivinsky et al., 1995*; *Kubo et al., 1993*; *Lesage et al., 1994*; *Lesage et al., 1995*). GIRK1/4 channels were expressed as previously described (*Touhara et al., 2016*) and cryo-EM samples were prepared in the presence of 0.5 mM C8-PIP$_2$. Data were collected and images processed as shown (*Figure 3—figure supplement 1*). 3D classification revealed both extended (26% particles) and docked (22% particles) conformations (*Figure 3* and *Figure 3—figure supplement 1*). Three remaining classes, accounting for 52% of the data, were insufficiently resolved to determine details of the underlying conformations. The two resolved conformations exhibit features similar to those of GIRK2, with lengths of ~119 Å and ~112 Å for extended and docked conformations, respectively, a well-resolved linker in the docked conformation but not in the extended conformation, and a 35° difference in the rotation of the CTD with respect to the TMD (*Figure 3*). We conclude that GIRK1/4, like GIRK2, can adopt the extended and docked conformations.

## Conformational changes in the PIP$_2$ and G$_{\beta\gamma}$ binding sites on GIRK2

Using all particles classified as 'docked' in the titration dataset, we reconstructed a best map of the GIRK2 channel with four bound PIP$_2$ molecules (CryoEM$^{PIP2}$) at a resolution of 3.3 Å (*Figure 4A*, *Figure 4—figure supplements 1* and *2*). The TMD-CTD linker forms a well-resolved helix in contrast to a flexible loop in the CryoEM$^{Apo}$ structure. This conformational change positions Lys199 and Lys200, along with Lys194 from the inner helix and Lys90 from the outer helix, to form electrostatic interactions with the PIP$_2$ molecule (*Figure 4B*). In addition, the 6 Å translation and 35° rotation of the CTD associated with PIP$_2$ binding brings Lys64 (from the N-terminus) near enough to PIP$_2$ to engage its 4'-phosphate (*Figure 4B and C* and *Figure 4—figure supplement 3*). The sidechains of positively charged residues that coordinate PIP$_2$ are disordered in the CryoEM$^{Apo}$ structure. Thus, PIP$_2$ binding stabilizes the docked conformation through interactions between the 4',5'-phosphate-substituted inositol head group of PIP$_2$ and Lys64 (with the 4' phosphate) and Lys194, Lys199, Lys200 (with the 5' phosphate). An analogous constellation of charge-pair stabilization is observed in the PIP$_2$ bound form of Kir2.2 (*Hansen et al., 2011*; *Tao et al., 2009*).

Another conformational change associated with PIP$_2$ binding occurs on the side of the CTD, involving the βL-βM loop, as shown (*Figure 4D and E*). The βL-βM loop does not make direct contact with the PIP$_2$ binding site but apparently is allosterically coupled to it. The possible importance of the βL-βM loop conformational change is implied through superposition of CryoEM$^{Apo}$ CTD and CryoEM$^{PIP2}$ CTD with the CTD from the crystal structure of GIRK2 in complex with G$_{\beta\gamma}$ (Crystal$^{GIRK-G\beta\gamma}$). The βL-βM loop is located on the surface to which G$_{\beta\gamma}$ binds. In CryoEM$^{PIP2}$, the βL-βM loop adopts the position observed in the G$_{\beta\gamma}$ complex. Thus, it would appear that PIP$_2$ binding to GIRK2 pre-configures the G$_{\beta\gamma}$ binding surface on the CTD into a receptive conformation.

## Discussion

The main conclusion of this study is that GIRK channels – GIRK2 as well as GIRK1/4 – can adopt extended and docked conformations. With GIRK2 we show that the docked conformation is favored in the presence of PIP$_2$. We think that crystal contacts in the previous Crystal$^{Apo}$ structure pushed the CTD and TMD into close proximity (*Whorton and MacKinnon, 2011*). It seems likely that PIP$_2$ in

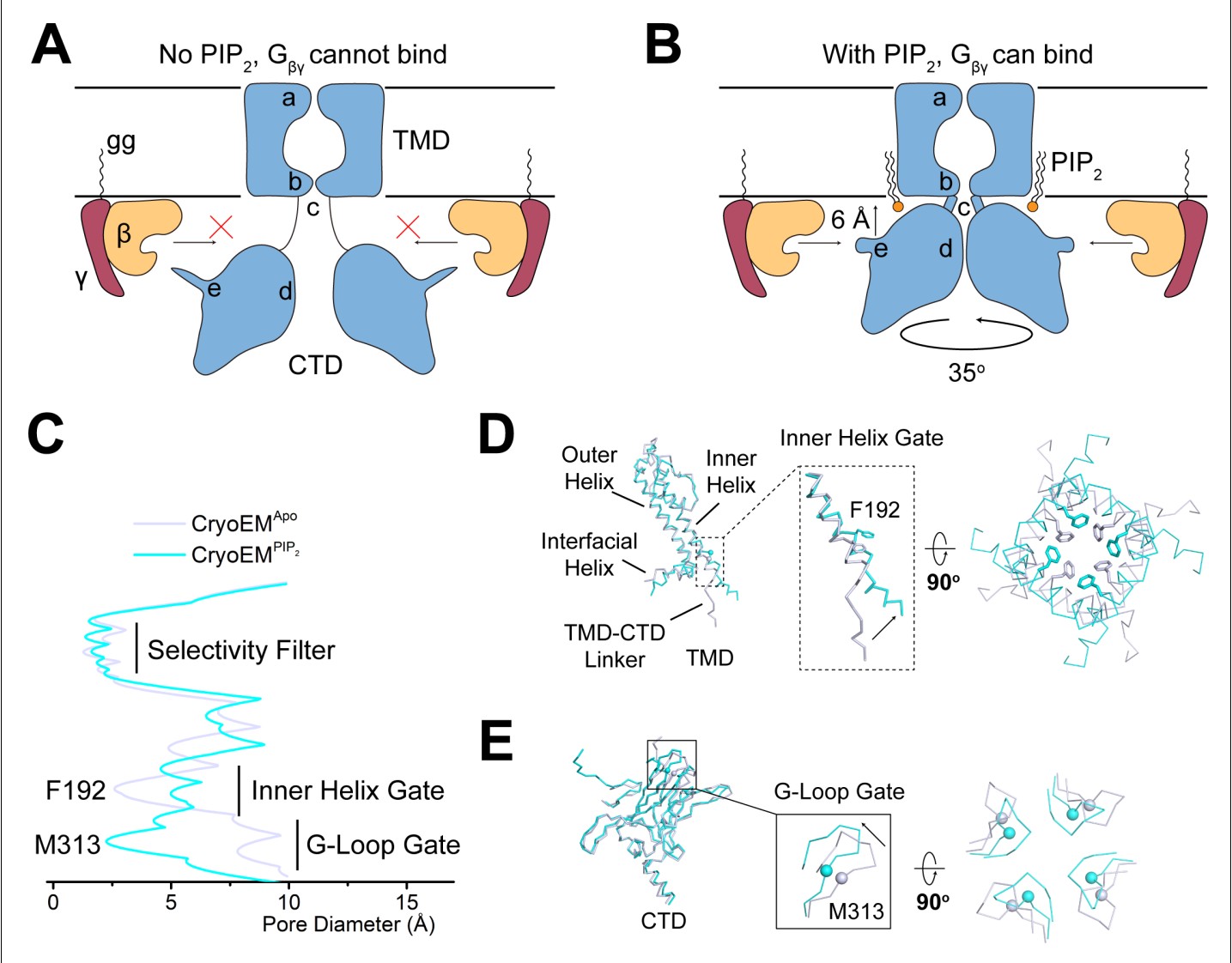

**Figure 5.** PIP₂ serves as an allosteric regulator to permit G$_{βγ}$ binding. (**A and B**) A cartoon depiction of PIP₂ regulation of GIRK channels. The blue shape depicts the GIRK channel. a, b, c, d, and e indicate the selectivity filter, inner helix gate, TMD-CTD linker, the G-loop gate, and the βL-βM loop, respectively. Circular arrow indicates the rotation about the pore axis with respect to the TMD and perpendicular arrow indicates CTD translation upon PIP₂ binding. The 'gg' label represents the geranylgeranyl lipid modification at the C terminus of G$_γ$. In the absence of PIP₂ (**A**), GIRK2 channel adopts the extended conformation and is not positioned for G$_{βγ}$ binding. Upon PIP₂ binding (**B**), the GIRK2 channel transits to the docked conformation, allowing G$_{βγ}$ binding to occur. (**C**) Plot of pore diameter (between van der Waals surfaces, calculated with Hole) for the extended (gray) and docked (cyan) structures. (**D and E**) Superposition of the channel TMD from the extended (gray) and docked (cyan) structures. Conformational changes at the inner helix gate (**D**) and G-loop gate (**E**) are boxed and zoomed-in details are shown. Sidechains of the inner helix gate-forming residue Phe192 are shown as sticks. Cα atoms of the G-loop gate constriction residue Met313 are shown as spheres.

GIRK channels, as previously proposed for Kir2.2 channels (*Hansen et al., 2011*; *Tao et al., 2009*), mediates the docking of the CTD onto the TMD. Electrostatic interactions between the anionic headgroup of PIP₂ and cationic amino acid sidechains on the channel serve to tether the CTD to the TMD. Many of these electrostatic interactions are conserved in GIRK and Kir2.2. A secondary conformational change occurs on the side of the CTD corresponding to the G$_{βγ}$ binding surface. This change configures the surface ready to bind G$_{βγ}$.

These effects of PIP₂ on GIRK2 are summarized in a cartoon (*Figure 5A and B*). By tethering the CTD to the TMD, PIP₂ brings the CTD close to the membrane surface where it can be reached by G$_{βγ}$, which is held at the membrane surface by its covalent attachment to a lipid tail. Coincidentally,

the binding surface for $G_{\beta\gamma}$ on the CTD adopts a permissive conformation, allowing $G_{\beta\gamma}$ to bind. This proposed mechanism is consistent with the demonstration that $G_{\beta\gamma}$ is unable to open GIRK2 in the absence of $PIP_2$ (*Wang et al., 2014*): without $PIP_2$, $G_{\beta\gamma}$ is unable to access its binding site on GIRK2.

How do the $PIP_2$-induced conformational changes help to open the pore? GIRK channels have two gates along their ion conduction pore: an inner helix gate in the TMD and a G-loop gate at the apex of the CTD (*Doyle et al., 1998*; *Jiang et al., 2002*; *Nishida et al., 2007*; *Pegan et al., 2005*; *Whorton and MacKinnon, 2011*; *Whorton and MacKinnon, 2013*). In the absence of $PIP_2$, the inner helix gate of GIRK2 is most tightly constricted at position Phe192 in the inner helix, which lines the pore on the intracellular side of the selectivity filter (*Figure 5C and D*). $PIP_2$ binding and docking of the CTD onto the TMD is associated with a modest change in the conformation of the inner helix, a repositioning of the sidechain of Phe192, and widening of the pore. The G-loop gate actually constricts when the CTD docks onto the TMD (*Figure 5C and E*). Thus, it would appear that when four $PIP_2$ molecules bind to GIRK2, the inner helix gate opens, or begins to open, but the pore remains closed owing to a constricted G-loop gate. This interpretation is consistent with functional data showing that $PIP_2$ is unable to open GIRK2 in the absence of $G_{\beta\gamma}$ (*Wang et al., 2014*) because the G-loop gate remains closed.

In summary, it appears that $PIP_2$ in GIRK channels enables $G_{\beta\gamma}$-mediated opening by bringing the CTD near the intracellular membrane surface and rendering the $G_{\beta\gamma}$ binding surface permissive for attachment.

# Materials and methods

## Key resources table

| Reagent type (species) or resource | Designation | Source or reference | Identifiers | Additional information |
|---|---|---|---|---|
| Gene (*Mus musculus* GIRK2) | GIRK2 | synthetic | | Synthesized at GeneWiz. |
| Gene (*Homo sapiens* GIRK1) | GIRK1 | synthetic | | Synthesized at GeneWiz. |
| Gene (*Homo sapiens* GIRK4) | GIRK4 | synthetic | | Synthesized at GeneWiz. |
| Strain, strain background (*Escherichia coli*) | DH10Bac | ThermoFisher | 10361012 | |
| Recombinant DNA reagent | pPICZ-GIRK2 | https://doi.org/10.1016/j.cell.2011.07.046 | | Maintained at the Mackinnon lab |
| Recombinant DNA reagent | GIRK1-His10-pEG BacMam | https://doi.org/10.7554/eLife.15750.001 | | Maintained at the Mackinnon lab |
| Recombinant DNA reagent | GIRK4-1D4-pEG BacMam | https://doi.org/10.7554/eLife.15750.001 | | Maintained at the Mackinnon lab |
| Cell line (*Pichia pastoris*) | SMD1163 | Invitrogen | C17500 | |
| Cell line (*Spodoptera frugiperda*) | Sf9 | ATCC | Cat# CRL-1711 | |
| Cell line (*Homo sapiens*) | HEK293S GnTI⁻ | ATCC | Cat# CRL-3022 | |

*Continued on next page*

*Continued*

| Reagent type (species) or resource | Designation | Source or reference | Identifiers | Additional information |
|---|---|---|---|---|
| Chemical compound, drug | SF-900 II SFM medium | GIBCO | Cat# 10902–088 | |
| Chemical compound, drug | L-Glutamine (100x) | GIBCO | Cat# 25030–081 | |
| Chemical compound, drug | Pen Strep | GIBCO | Cat# 15140–122 | |
| Chemical compound, drug | Grace's insect medium | GIBCO | Cat# 11605–094 | |
| Chemical compound, drug | Freestyle 293 medium | GIBCO | Cat# 12338–018 | |
| Chemical compound, drug | Fetal bovine serum | GIBCO | Cat# 16000–044 | |
| Chemical compound, drug | Cellfectin II reagent | Invitrogen | Cat# 10362100 | |
| Chemical compound, drug | Cholesteryl hemisuccinate (CHS) | Anatrace | CH210 | |
| Chemical compound, drug | n-Dodecyl-β-D-Maltopyranoside (DDM) | Anatrace | D310S | |
| Chemical compound, drug | n-Decyl-β-D-Maltopyranoside (DM) | Anatrace | D322S | |
| Chemical compound, drug | 1,2-dioctanoyl-sn-glycero-3-phospho-(1'-myo-inositol-4',5'-bisphosphate) (ammonium salt) (C8-PIP$_2$) | Avanti Polar Lipids | 850185P | |
| Chemical compound, drug | (1H, 1H, 2H, 2H-Perfluorooctyl) phosphocholine (FFC8) | Anatrace | F300F | |
| Commercial assay or kit | CNBr-activated Sepharose beads | GE Healthcare | Cat# 17-0430-01 | |
| Commercial assay or kit | Superdex 200 Increase 10/300 GL | GE Healthcare Life Sciences | 28990944 | |
| Commercial assay or kit | R1.2/1.3 400 mesh Au holey carbon grids | Quantifoil | 1210627 | |
| Commercial assay or kit | Superose 6 Increase 10/300 GL | GE Healthcare Life Sciences | 29091596 | |
| Software, algorithm | RELION 3.0 | https://doi.org/10.7554/eLife.42166.001 | http://www2.mrc-lmb.cam.ac.uk/relion | |

*Continued on next page*

*Continued*

| Reagent type (species) or resource | Designation | Source or reference | Identifiers | Additional information |
|---|---|---|---|---|
| Software, algorithm | RELION 3.1 | https://doi.org/10.1101/798066 | http://www2.mrc-lmb.cam.ac.uk/relion | |
| Software, algorithm | MotionCor2 | https://doi.org/10.1038/nmeth.4193 | http://msg.ucsf.edu/em/software/motioncor2.html | |
| Software, algorithm | Gctf 1.0.6 | https://doi.org/10.1016/j.jsb.2015.11.003 | https://www.mrc-lmb.cam.ac.uk/kzhang/Gctf/ | |
| Software, algorithm | CtfFind4.1.8 | https://doi.org/10.1016/j.jsb.2015.08.008 | http://grigoriefflab.janelia.org/ctffind4 | |
| Software, algorithm | Gautomatch | | https://www.mrc-lmb.cam.ac.uk/kzhang/Gautomatch/ | |
| Software, algorithm | CryoSPARC 2.4.0 | https://doi.org/10.7554/eLife.46057.001 | https://cryosparc.com/ | |
| Software, algorithm | Pyem | | https://github.com/asarnow/pyem | |
| Software, algorithm | COOT | https://doi.org/10.1107/S0907444910007493 | http://www2.mrc-lmb.cam.ac.uk/personal/ pemsley/coot | |
| Software, algorithm | PHENIX | https://doi.org/10.1107/S0907444909052925 | https://www.phenix-online.org | |
| Software, algorithm | Adobe Photoshop version 16.0.0 (for figure preparation) | Adobe Systems, Inc | | |
| Software, algorithm | GraphPad Prism version 8.0 | GraphPad Software | | |
| Software, algorithm | MacPyMOL: PyMOL v2.0 Enhanced for Mac OS X | Schrodinger LLC | https://pymol.org/edu/?q=educational/ | |
| Software, algorithm | Chimera | https://doi.org/10.1002/jcc.20084 | https://www.cgl.ucsf.edu/chimera/download.html | |
| Software, algorithm | Serial EM | https://doi.org/10.1016/j.jsb.2005.07.007 | http://bio3d.colorado.edu/SerialEM | |
| Software, algorithm | HOLE | https://doi.org/10.1016/S0263-7855(97)00009-X | http://www.holeprogram.org | |

## Protein expression and purification

Mouse GIRK2 (residues 52–380) was expressed in *Pichia pastoris* as previously described (*Whorton and MacKinnon, 2011*). Frozen cells were lysed in a mixer mill, and resuspended in 50 mM HEPES (pH 7.4), 150 mM KCl and protease inhibitor cocktail for 1 hr at 4°C. The resuspension was adjusted to pH 8.0, then 4% (w/v) n-decyl-β-D-maltopyranoside (DM) and 0.8% (w/v) cholesterol hemisuccinate (CHS) (or 4% n-dodecyl-β-D-maltopyranoside (DDM) and 0.8% CHS for the apo GIRK2 sample) were added to extract for 2 hr at 4°C. The mixture was centrifuged at 37,500 g for 30 min and the supernatant incubated with GFP nanobody-coupled CNBr-activated Sepharose resin (GE Healthcare) for 1–2 hr at 4°C (*Kubala et al., 2010*). The resin was subsequently washed with 10 column volumes of wash buffer (50 mM HEPES pH 7.4, 150 mM KCl, 0.2% DM and 0.04% CHS for structural titration samples, or 0.05% DDM and 0.01% CHS for apo GIRK2 sample). The washed resin

was incubated overnight with PreScission protease at a target protein to protease ratio of 40:1 (w:w) to cleave off GFP and release the protein from the resin. The protein was eluted with wash buffer, concentrated using an Amicon Ultra centrifugal filter (MWCO 100 kDa), and then injected onto a Superdex 200 increase 10/300 GL column (GE Healthcare) equilibrated with SEC buffer (20 mM Tris-HCl pH 7.5, 150 mM KCl, 10 mM DTT, 1 mM EDTA, 0.2% DM and 0.04% CHS for structural titration samples, or 0.05% DDM and 0.01% CHS for apo GIRK2 sample). Peak fractions corresponding to the GIRK2 tetramer were pooled and concentrated to 6–7 mg/ml using an Amicon Ultra centrifugal filter (MWCO 100 kDa).

Full-length human GIRK1 and GIRK4 gene**s** were cloned into a pEG BacMam vector, and co-expressed in HEK293S GnTI⁻ (ATCC CRL-3022) cells as previously described (*Touhara et al., 2016*). Cells were solubilized in 50 mM HEPES (pH 7.4), 150 mM KCl, 1.5% (w/v) DDM, 0.3% (w/v) CHS, and protease inhibitor cocktail. Two hours after solubilization, lysed cells were centrifuged at 37,500 g for 30 min and supernatant was incubated with Talon metal affinity resin (Clontech Laboratories, Inch. Mountain View, CA) for 1 hr at 4°C with gentle mixing. The resin was washed in batch with five column volumes of buffer A (50 mM HEPES pH 7.0, 150 mM KCl, 0.05% [w/v] DDM, 0.01% [w/v] CHS), then loaded onto a column and further washed with five column volumes of buffer A + 10 mM imidazole. The protein was then eluted with buffer A + 200 mM imidazole. The peak fraction was collected and incubated with the 1D4 affinity resin for 1 hr at 4°C with gentle mixing. The resin was loaded onto a column and washed with buffer A. Five mM DTT and 1 mM EDTA were added and eGFP and affinity tags were cut with PreScission protease overnight at 4°C. The cleaved protein was then concentrated to run on a Superose 6 10/300 GL gel filtration column in 20 mM Tris-HCl (pH 7.5), 150 mM KCl, 100 mM NaCl, 0.025% (w/v) DDM, 0.005% (w/v) CHS, 10 mM DTT, and 1 mM EDTA. Protein was finally concentrated to ~5 mg/ml using an Amicon Ultra centrifugal filter (MWCO 100 kDa).

## Cryo-EM sample preparation and data collection

For the apo GIRK2 sample, purified GIRK2 at a concentration of 6–7 mg/ml was mixed with Fluorinated Fos-Choline-8 (FFC8) (Anatrace) stock at 29 mM to a final concentration of 2.9 mM immediately prior to application of 3.5 µL of the mixture onto a glow-discharged Quantifoil R1.2/1.3 400 mesh Au grid (Quantifoil), blotted for 4 s at room temperature (RT) with a blotting force of 2–4 and plunge-frozen in liquid ethane using a Vitrobot Mark IV (FEI).

For the structural titration samples, purified GIRK2 at a concentration of 6–7 mg/ml was mixed with C8-PIP$_2$ stock at 10 mM to a final concentration of 0, 0.25, 0.5, 0.75, 0.875 or 1 mM. In an earlier study (*Wang et al., 2014*), we showed that C8-PIP$_2$ activates GIRK2 channels in lipid membranes with an activation constant ~15 µM and Hill coefficient n ~ 3. C8-PIP$_2$ partitions into membranes and thus the local concentration near the channel is unknown. We chose the higher concentration range in the structural study because the channels are in detergent micelles, where local concentrations of PIP$_2$, compared to studies with lipid membranes, are expected to be lower. The mixtures were further mixed with FFC8 stock to a final concentration of 2.9 mM, and 3.5 µL aliquots of the protein were pipetted onto glow-discharged Quantifoil R1.2/1.3 400 mesh Au grids and blotted with the same settings as for apo GIRK2 using a Vitrobot Mark IV.

For GIRK1/4 samples, purified GIRK1/4 at a concentration of ~5 mg/ml was mixed with C8-PIP$_2$ stock at 10 mM to the final concentration of 0.5 mM. 3.5 µL aliquots of the protein were then pipetted onto glow-discharged Quantifoil R1.2/1.3 400 mesh Au grids and blotted for 1 s at RT with a blotting force of 1 and plunge-frozen in liquid ethane using a Vitrobot Mark IV.

Cryo-EM data were collected on a 300-kV Titan Krios electron microscope (Thermo Fisher Scientific) equipped with a K3 Summit (apo GIRK2 sample) or K2 Summit (other samples) direct electron detector in super-resolution mode. After binning over 2 × 2 pixels, the calibrated pixel size was 0.86 Å, 1 Å and 1.03 Å for apo GIRK2, structural titration samples and GIRK1/4, respectively. For the apo GIRK2 sample, exposures of 3 s were dose-fractionated into 80 frames with a dose rate of 25.5 electrons per pixel per second, resulting in a total dose of 103.3 electrons per Å$^2$. For structural titration samples and GIRK1/4 samples, exposures of 10 s were dose-fractionated into 50 frames with a dose rate of 8 electrons per pixel per second, resulting in a total dose of 80 (structural titration samples) or 75.4 (GIRK1/4 sample) electrons per Å$^2$, respectively. Cryo-EM data collection statistics are summarized in *Table 1*.

**Table 1.** Cryo-EM data collection and refinement statistics, related to **Figures 1**, **2** and **4**.

| | GIRK2[Extended] | GIRK2[Docked] | GIRK1/4[Extended] | GIRK1/4[Docked] |
|---|---|---|---|---|
| EMDB ID | EMD-22199 | EMD-22200 | EMD-22201 | EMD-22202 |
| PDB ID | 6XIS | 6XIT | | |
| **Data collection** | | | | |
| Microscope | Titan Krios | | | |
| Detector | K3 summit | K2 summit | | |
| Voltage (kV) | 300 | | | |
| Pixel size (Å) | 0.43 | 0.5 | 0.515 | |
| Total electron exposure (e$^-$/Å$^2$) | 103.3 | 80.0 | 75.4 | |
| Defocus range (µm) | 1.0 to 3.0 | 1.5 to 2.5 | 1.5 to 3.5 | |
| Micrographs collected | 2103 | 11,349 | 3415 | |
| **Reconstruction** | | | | |
| Final particle images | 112,517 | 155,128 | 57,644 | 48,757 |
| Pixel size (Å) | 1.29 | 1 | 1.03 | 1.03 |
| Box size (pixels) | 256 | 400 | 256 | 256 |
| Resolution (Å) (FSC = 0.143) | 3.9 | 3.3 | 7.9 | 4.6 |
| Map Sharpening B-factor (Å$^2$) | −26 | −12 | - | −192 |
| **Model composition** | | | | |
| Non-hydrogen atoms | 9460 | 10,252 | | |
| Protein residues | 1240 | 13,08 | | |
| Ligands | 0 | 4 | | |
| Metals | 0 | 3 | | |
| **Refinement** | | | | |
| Model-to-map CC (mask) | 0.62 | 0.72 | | |
| Model-to-map CC (volume) | 0.64 | 0.75 | | |
| **R.m.s deviations** | | | | |
| Bond length (Å) | 0.006 | 0.009 | | |
| Bond angles (°) | 1.3 | 1.3 | | |
| **Validation** | | | | |
| MolProbity score | 2.00 | 1.80 | | |
| Clash score | 8.55 | 9.91 | | |
| **Ramachandran plot** | | | | |
| Outliers (%) | 0 | 0 | | |
| Allowed (%) | 1.7 | 4.0 | | |
| Favored (%) | 98.3 | 96.0 | | |
| Rotamer outliers (%) | 1.40 | 0.74 | | |
| C-beta deviations (%) | 0 | 0 | | |

## Cryo-EM data processing

Image processing was performed in both CryoSPARC-2.9.0 (Structura Biotechnology) (*Punjani et al., 2017*) and RELION (*Scheres, 2012*; *Zivanov et al., 2018*; *Zivanov et al., 2020*). All movie frames were corrected with a gain reference collected during the same EM session, and specimen movement was corrected using MotionCorr2 (*Zheng et al., 2017*) with dose weighting. The contrast transfer function (CTF) parameters were estimated using Gctf-1.0.6 (*Zhang, 2016*). Images showing

substantial ice contamination, abnormal background, thick ice, low contrast or poor Thon rings were discarded.

For structure determination of the apo GIRK2, 2049 of 2103 micrographs were selected for further processing. Particles were picked with the Laplacian-of-Gaussian auto-picking implemented in RELION-3 without templates. 502,731 auto-picked particles were extracted into $384 \times 384$ pixel images. The particle images were binned 1.5 times and subjected to Ab-initio reconstruction in CryoSPARC-2.9.0, specifying four output classes. The best class, including 216,085 particles were selected for homogeneous refinement with C4 symmetry, which yielded a map at 5.6 Å resolution. The particles were transferred back to RELION-3 using the pyem package (https://github.com/asar-now/pyem), re-extracted into $320 \times 320$ pixel images, binned 1.5 times and further refined with C4 symmetry after Bayesian polishing and CTF refinement, generating an improved map at 4.3 Å resolution. Focused refinement was performed on the CTD region, followed by the focused 3D classification on the TMD region without image alignment similar to a previous report (*Lee and MacKinnon, 2018*), revealing significant heterogeneity at the TMD region. Finally, 112,571 particles with fine features of TMD were selected and subjected to Non-uniform refinement and local refinement in CryoSPARC-2.9.0, which yielded the final map at 3.9 Å resolution.

For the structural titration of the GIRK2 channel, roughly 90–95% of micrographs from each $PIP_2$ concentration dataset were selected for further processing. Different views from 2D averages of the apo GIRK2 dataset were selected as templates for particle picking using Gautomatch (https://www.mrc-lmb.cam.ac.uk/kzhang/Gautomatch/). Auto-picked particles for each $PIP_2$ concentration were extracted into $384 \times 384$ pixel images, binned 1.5 times, and subjected to Ab-initio reconstruction in CryoSPARC-2.9.0, specifying four output classes (*Punjani et al., 2017*). Non-protein particles were removed, resulting in 99,381, 109,462, 105,388, 123,649, 99,619 and 105,708 particle images for the 0, 0.25, 0.5, 0.75, 0.875 and 1 mM $PIP_2$ datasets, respectively. After manual inspection, all these particles were combined (a total of 643,657) and transferred back to RELION-3. The particles were re-extracted into $320 \times 320$ pixel images, binned 1.5 times and subjected to 3D refinement in RELION-3 with the density map of apo GIRK2 as reference (low-pass filtered to 60 Å). Subsequently, the angular and translational parameters determined using 3D auto-refine were fixed throughout 25 cycles of 3D classification, specifying five classes in RELION-3 with the map generated by 3D auto-refine low-pass filtered to 60 Å serving as the initial model. Manual inspection identified one class with characteristics of the docked class and contributions of all six datasets to this 3D class were determined using identifiers uniquely associated with each particle during particle extraction. To ensure the reproducibility of the 3D classification algorithm, the 643,657 particles were subjected to five independent runs of 3D auto-refine and 3D classification.

For structure determination of GIRK2 with $PIP_2$ bound, the 155,128 particles corresponding to the docked class in the structural titration dataset was refined applying C4 symmetry using the Non-uniform refinement algorithm in CryoSPARC-2.9., resulting in a map at 4.6 Å resolution. The refined particles were then transferred back to RELION, followed by CTF refinement and Bayesian polishing, which improved the map to 4.0 Å. Particles were then re-extracted into $400 \times 400$ pixel images and subjected to multiple runs of CTF refinement in RELION-3.1 as described (*Scheres, 2019*; *Zivanov et al., 2020*). Further 3D refinement with C4 symmetry in RELION-3.1 yielded a final map at 3.3 Å resolution.

For the GIRK1/4 channel dataset with 0.5 mM C8-$PIP_2$, 3377 of 3415 micrographs were selected for further processing. Different views of 2D averages from the apo GIRK2 dataset were selected as templates and Gautomatch was used for particle picking. The auto-picked 554,081 particles were extracted into $256 \times 256$ pixel images, binned two times and subjected to Ab-initio reconstruction in CryoSPARC-2.9.0, specifying four output classes. The best of the four classes, including 221,633 particles, were selected for homogeneous refinement applying C2 symmetry, which yielded a map at 7.7 Å resolution. The particles were transferred back to RELION-3 using the pyem package with angular and translational parameters and re-extracted into $256 \times 256$ pixel images. 3D classification was performed requesting five classes without image alignment, using the refined map from CryoSPARC low-pass filtered to 60 Å as the initial model. Two out of five classes showed typical features of docked and extended conformations, accounting for 22% (48,757) and 26% (57,644) of the particles, respectively. These two classes were processed separately using the same approach. Specifically, per-particle Defocus-U and Defocus-V values were first determined in the local-fitting mode of Gctf-1.0.6. 3D refinement with C2 symmetry yielded the final map of the extended class at 7.9 Å.

For the docked class, local CTF estimation yielded a map at 6.1 Å resolution and subsequent Bayesian polishing and CTF refinement with C2 symmetry further improved the map to 4.6 Å resolution.

We chose C2 symmetry in the analysis of GIRK1/4 images based on the empirical observation that it yielded a better map than refinement without symmetry. The resolution of the GIRK1/4 data precluded distinction between the GIRK1 and GIRK4 subunits; however, the improved refinement by application of C2 symmetry implies that these subunits might alternate in their positions around the pore's central axis.

### Model building and refinement

For the apo GIRK2 structure (extended conformation), models of the CTD and TMD region from the crystal structure of GIRK2 channel monomer without PIP$_2$ (PDB: 3SYO) were placed into the density using UCSF Chimera (*Pettersen et al., 2004*). Then the two regions were manually connected in Coot and rounds of real-space refinement were performed in Phenix with secondary structure restraints (*Adams et al., 2010*). The refined channel monomer was then copied and fit into corresponding density of the other three subunits using the Jiggle-fit and chain-refine command of Coot 0.9 in the CCPEM suite (*Burnley et al., 2017*; *Wood et al., 2015*) to generate the channel tetramer. Finally, several iterative cycles of refinement using the phenix.real_space_refine in PHENIX with secondary structure and NCS restraints and manual adjustments in COOT yielded the final model for the apo structure of the GIRK2 channel.

For the structure of GIRK2 with PIP$_2$ (docked conformation), models of channel monomer and the PIP$_2$ molecule (PDB: 3SYA) were placed into density using UCSF Chimera (*Pettersen et al., 2004*) and several rounds of real-space refinement were performed in Phenix with secondary structure restraints (*Adams et al., 2010*). The channel tetramer was generated using Coot 0.9 in a similar manner to the apo GIRK2 structure. Finally, iterative refinement cycles using the phenix.real_space_refine in PHENIX with secondary structure and NCS restraints and manual adjustments in COOT yielded the final model for the docked conformation of GIRK2 channel with 4 PIP$_2$ molecules bound.

Refinement statistics are summarized in *Table 1*. For model validation, the final model for each map was refined against one of the half maps (half map 1). FSC curves were then calculated between the refined model and half map 1 (work), half map 2 (free) as well as the combined full map. Local resolutions were estimated using Relion3 (*Zivanov et al., 2018*).

### Structural analysis

Structural alignment and figures were made in UCSF Chimera (*Pettersen et al., 2004*) and PyMOL (www.pymol.org). Pore diameter between van der Waals surfaces for the extended and docked structures of the GIRK2 channel was calculated with Hole (www.holeprogram.org).

## Acknowledgements

We thank Mark Ebrahim and Johanna Sotiris at the Evelyn Gruss Lipper Cryo-EM Resource Center at Rockefeller University for assistance in data collection; Dr. Chia-Hsueh Lee (St. Jude Children's Research Hospital) for critical reading of the manuscript and suggestions for image analysis; Dr. Yixiao Zhang and Dr. Hiroshi Suzuki (Rockefeller University) for advice and help on data collection; Dr. Richard Hite (Memorial Sloan Kettering Cancer Center) for advice on the structural titration analysis; and members of the MacKinnon lab and Chen lab (Rockefeller University) for assistance. This work was supported in part by GM43949. RM is an investigator in the Howard Hughes Medical Institute.

## Additional information

### Funding

| Funder | Grant reference number | Author |
| --- | --- | --- |
| National Institutes of Health | GM43949 | Roderick MacKinnon |
| Howard Hughes Medical Institute | | Roderick MacKinnon |

The funders had no role in study design, data collection and interpretation, or the decision to submit the work for publication.

## Author contributions

Yiming Niu, Xiao Tao, Data curation, Software, Validation, Visualization, Methodology, Writing - original draft, Writing - review and editing; Kouki K Touhara, Data curation, Methodology, Writing - review and editing; Roderick MacKinnon, Conceptualization, Resources, Data curation, Supervision, Funding acquisition, Validation, Investigation, Visualization, Writing - original draft, Project administration, Writing - review and editing

## Author ORCIDs

Yiming Niu (iD) https://orcid.org/0000-0002-5683-1781
Xiao Tao (iD) https://orcid.org/0000-0002-9381-7903
Kouki K Touhara (iD) http://orcid.org/0000-0003-3167-9784
Roderick MacKinnon (iD) https://orcid.org/0000-0001-7605-4679

## Decision letter and Author response

Decision letter https://doi.org/10.7554/eLife.60552.sa1
Author response https://doi.org/10.7554/eLife.60552.sa2

# Additional files

## Supplementary files

• Transparent reporting form

## Data availability

The B-factor sharpened 3D cryo-EM density map and atomic coordinates of GIRK2 in the extended conformation (GIRK2Extended) and GIRK2 in the docked conformation with PIP2 (GIRK2Docked) have been deposited in the Worldwide Protein Data Bank (wwPDB) under accession number EMD-22199 and 6XIS, EMD-22200 and 6XIT, respectively. The B-factor sharpened 3D cryo-EM density map of GIRK1/4 in the extended conformation (GIRK1/4Extended) and docked conformation with PIP2 (GIRK1/4Docked) have been deposited in the Worldwide Protein Data Bank (wwPDB) under accession number EMD-22201 and EMD-22202, respectively.

The following datasets were generated:

| Author(s) | Year | Dataset title | Dataset URL | Database and Identifier |
|---|---|---|---|---|
| Niu Y, Tao X, MacKinnon R | 2020 | Cryo-EM structure of the G protein-gated inward rectifier K+ channel GIRK2 (Kir3.2) in complex with PIP2 | http://www.rcsb.org/structure/6XIT | RCSB Protein Data Bank, 6XIT |
| Niu Y, Tao X, MacKinnon R | 2020 | Cryo-EM structure of the G protein-gated inward rectifier K+ channel GIRK2 (Kir3.2) in complex with PIP2 | https://www.ebi.ac.uk/pdbe/entry/emdb/EMD-22200 | Electron Microscopy Data Bank, EMD-22200 |
| Niu Y, Tao X, MacKinnon R | 2020 | Cryo-EM structure of the G protein-gated inward rectifier K+ channel GIRK2 (Kir3.2) in apo form | http://www.rcsb.org/structure/6XIS | RCSB Protein Data Bank, 6XIS |
| Niu Y, Tao X, MacKinnon R | 2020 | Cryo-EM structure of the G protein-gated inward rectifier K+ channel GIRK2 (Kir3.2) in apo form | https://www.ebi.ac.uk/pdbe/entry/emdb/EMD-22199 | Electron Microscopy Data Bank, EMD-22199 |
| Niu Y, Tao X, MacKinnon R | 2020 | Cryo-EM structure of the G protein-gated inward rectifier K+ channel GIRK1/4 (Kir3.1/Kir3.4) in apo form | https://www.ebi.ac.uk/pdbe/entry/emdb/EMD-22201 | Electron Microscopy Data Bank, EMD-22201 |
| Niu Y, Tao X, MacKinnon R | 2020 | Cryo-EM structure of the G protein-gated inward rectifier K+ channel GIRK1/4 (Kir3.1/Kir3.4) in complex with bound PIP2 | https://www.ebi.ac.uk/pdbe/entry/emdb/EMD-22202 | Electron Microscopy Data Bank, EMD-22202 |

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
