## [Decision Letter]

**Acceptance summary:**

This paper presents structural data showing that the signaling lipid PIP_2_ biases the structure of a G-protein regulated potassium channel (GIRK2) towards a "docked" conformation. The authors present a model whereby this PIP_2_-mediated docking allows binding of the G-protein subunits. The data and model therefore explain why both PIP_2_ and G-proteins are necessary for the opening of GIRK2 channels.

**Decision letter after peer review:**

Thank you for submitting your article "Cryo-EM analysis of PIP_2_ regulation in GIRK channels" for consideration by *eLife*. Your article has been reviewed by three peer reviewers, and the evaluation has been overseen by Merritt Maduke as the Reviewing Editor and Kenton Swartz as the Senior Editor. The following individuals involved in review of your submission have agreed to reveal their identity: Bruce P Bean (Reviewer #1); Sudha Chakrapani (Reviewer #2); Ryan E Hibbs (Reviewer #3).

The reviewers have discussed the reviews with one another and the Reviewing Editor has drafted this decision to help you prepare a revised submission.

Summary:

This short paper presents cryo-EM data focused on structural elucidation of GIRK activation by PIP_2_. GIRK channels are unique among inward rectifiers in that they require the binding of both PIP_2_ and G_βγ_ proteins for activation, potentially affecting two gates, the inner gate at helix bundle and the G loop gate, respectively. The Mackinnon group previously determined structures of GIRK2 (in the absence and presence of PIP_2_) and a GIRK2-G_βγ_ complex by X-ray crystallography. While providing high-resolution structural insights into the architecture and G-protein activation, these structures did not reveal conformational changes that may underlie PIP_2_ regulation. Particularly, the transmembrane domain (TMD) and cytoplasmic domain (CTD) were tightly juxtaposed even in the absence PIP_2_. In addition, both the G-loop gate and the inner helix gate remained closed in the PIP_2_-bound structure. These results raised questions because PIP_2_ binding to GIRK2 did not result in the kind of conformational change observed in another inward rectifying K^+^ channel, Kir2.2.

Here, the authors report structures of neuronal GIRK2 in the apo and PIP2 bounds forms determined by cryo-EM. C8-PIP_2_ was titrated into GIRK2 samples, which enabled determination of cryo-EM structures of GIRK2 in docked (CTD and TMD tightly juxtaposed) and extended conformations. The docked conformation is progressively populated with increasing concentration of PIP_2_. The authors also report broadly similar findings for cardiac GIRK1/4 heteromeric channels.

Overall, the study is clearly written and logical and resolves a mystery for GIRK2- now we know it can adopt an extended conformation in the presence of PIP_2_. Further, the study demonstrates conservation in mechanism across several Kir members, as the docked vs. undocked conformational changes are similar to those observed with Kir2.2. Finally, the authors present a model whereby this PIP_2_-mediated docking allows binding of G_βγ_ to the channels. The data and model therefore explain why both PIP_2_ and G_βγ_ are necessary for the opening of GIRK channels.

Essential revisions:

1) The GIRK2 extended-conformation map quality is poor in some important regions. This map contains ~no reliable density for a slab between the CTD and TMD, however linkers and loops are modeled here. For example, 314-319 do not fit the map and 197-201 are built into very noisy density. Some regions of this map are quite good, and many are very noisy, consistent with Figure 1—figure supplement 1D showing a drop in FSC to ~0.5 at around 7A resolution. The Materials and methods describe this model being built through docking of earlier structures for the two halves followed by refinement (sounds fine). We ask that the authors clarify regions in which ambiguity is present in the Materials and methods section (or in the main text if they prefer), and describe the basis for modeling of the linkers (314-319 and 197-201) in particular. Please take a look also at 186-196. The helix appears to unwind in a couple of places (and the map is a bit ambiguous), and the important gate-keeping residue Phe 192 is modeled such that it pushes the backbone out of density. Maybe this is correct- it is unclear in the absence of more information.

2) Why were the GIRK1/4 data processed in C2? Is the subunit arrangement defined (understood) such that C2 is appropriate? Were C4 and C1 refinements or classifications attempted? Is pseudosymmetry is a concern? Are there classes with partial PIP_2_ occupancy?

3) The Materials and methods section doesn't describe how GIRK1/4 particle alignment was done. What structural features differ between 1 and 4 that aid alignment?

4) There are no functional assays shown in the manuscript, particularly to corroborate the shift in equilibrium population for the concentration range seen in cryo-EM. The authors may want to comment on why the docked conformation only constitutes ~30% of the total particles even at 1 mM PIP_2_. Are there other lipids such as cholesterol required?

5) Subsection “GIRK2 conformation as a function of PIP_2_ concentration”: How was the range of [C8-PIP_2_] chosen, and can the results of the titration be related to physiological experiments?

6) An interesting result is that even with no added PIP_2_, about 8% of the GIRK channels are in the docked conformation. If the authors have any thoughts about this it would probably be of interest to readers. Does this mean that there is a chance that some small fraction of channels could be bound by G_βγ_ even in the absence of PIP_2_? Or is there any chance that there is endogenous PIP_2_ in the expression system that can be retained during protein purification?

7) The figures for the conformational changes could be better with more detailed labeling (Figures 4 and 5).

8) What is the contour levels for the densities in Figure 1—figure supplement 2 and Figure 4—figure supplement 2?

9) Crystal packing is suggested to be responsible for the absence of obtaining an extended conformation previously. Is the preparation for the cryo-EM structural analysis being in detergent of any concern to the authors? The similarities to what has been seen with Kir2.2 provide some confidence.

10) The GIRK2 docked map and model look fine; PIP_2_ tails are not well resolved, which is worth mentioning perhaps in the Materials and methods.

---

## [Author Response]

Essential revisions:1) The GIRK2 extended-conformation map quality is poor in some important regions. This map contains ~no reliable density for a slab between the CTD and TMD, however linkers and loops are modeled here. For example, 314-319 do not fit the map and 197-201 are built into very noisy density. Some regions of this map are quite good, and many are very noisy, consistent with Figure 1—figure supplement 1D showing a drop in FSC to ~0.5 at around 7A resolution. The Materials and methods describe this model being built through docking of earlier structures for the two halves followed by refinement (sounds fine). We ask that the authors clarify regions in which ambiguity is present in the Materials and methods section (or in the main text if they prefer), and describe the basis for modeling of the linkers (314-319 and 197-201) in particular. Please take a look also at 186-196. The helix appears to unwind in a couple of places (and the map is a bit ambiguous), and the important gate-keeping residue Phe 192 is modeled such that it pushes the backbone out of density. Maybe this is correct- it is unclear in the absence of more information.

The density for 197-201 is essentially nonexistent and therefore we have removed these residues from the model (updated in Table 1) and changed wording in the text to reflect the absence of density for this region. We have also deleted this region in Figure 1, Figure 4, and Figure 1—figure supplement 3 to reflect such changes. However, this does not change our conclusions regarding separation of the CTD and TMD.

Density for 314-319 is continuous. We have further refined the atomic model for a better fit of the density. While there is uncertainty in the build, with restraints in model refinement weighted towards protein chemistry (as should be the case when refining a structure at this resolution), we think the model is the most objective molecular representation of the data.

Residues 186-196 do not appear to deviate from an α-helix and therefore we have built it as such.

2) Why were the GIRK1/4 data processed in C2? Is the subunit arrangement defined (understood) such that C2 is appropriate? Were C4 and C1 refinements or classifications attempted? Is pseudosymmetry is a concern? Are there classes with partial PIP_2_ occupancy?

The GIRK1/4 data were processed under C2 constraint because this yielded a better map. We added a new paragraph addressing this issue in the Materials and methods subsection “Cryo-EM data processing”. Occupancy of PIP_2_ is indeterminant at the resolution of this study.

3) The Materials and methods section doesn't describe how GIRK1/4 particle alignment was done. What structural features differ between 1 and 4 that aid alignment?

A more complete description of particle alignment is given in the Materials and methods subsection “Cryo-EM data processing”.

4) There are no functional assays shown in the manuscript, particularly to corroborate the shift in equilibrium population for the concentration range seen in cryo-EM. The authors may want to comment on why the docked conformation only constitutes ~30% of the total particles even at 1 mM PIP_2_. Are there other lipids such as cholesterol required?

In prior studies from our lab, published in *eLife* (Wang et al., 2014), we presented detailed analyses of PIP_2_ activation of GIRK channels. These previous data are now discussed in a new paragraph in the Results subsection “GIRK2 conformation as a function of PIP_2_ concentration” and in the Materials and methods subsection “Cryo-EM sample preparation and data collection”.

5) Subsection “GIRK2 conformation as a function of PIP_2_ concentration”: How was the range of [C8-PIP_2_] chosen, and can the results of the titration be related to physiological experiments?

This issue is addressed in the Materials and methods subsection “Cryo-EM sample preparation and data collection”.

6) An interesting result is that even with no added PIP_2_, about 8% of the GIRK channels are in the docked conformation. If the authors have any thoughts about this it would probably be of interest to readers. Does this mean that there is a chance that some small fraction of channels could be bound by G_βγ_ even in the absence of PIP_2_? Or is there any chance that there is endogenous PIP_2_ in the expression system that can be retained during protein purification?

We now address this point in a new paragraph in the Materials and methods subsection “Cryo-EM sample preparation and data collection”.

7) The figures for the conformational changes could be better with more detailed labeling (Figures 4 and 5).

We very much like these figures as they are.

8) What is the contour levels for the densities in Figure 1—figure supplement 2 and Figure 4—figure supplement 2?

The contour levels are now stated in the corresponding figure legends.

9) Crystal packing is suggested to be responsible for the absence of obtaining an extended conformation previously. Is the preparation for the cryo-EM structural analysis being in detergent of any concern to the authors? The similarities to what has been seen with Kir2.2 provide some confidence.

The distinction between detergent and lipid membranes in this study is important through the unique environmental influences on C8-PIP_2_ partitioning (and thus, local concentration). This is now discussed in the Materials and methods subsection “Cryo-EM sample preparation and data collection”.

10) The GIRK2 docked map and model look fine; PIP_2_ tails are not well resolved, which is worth mentioning perhaps in the Materials and methods.

Partial resolution of C8-PIP_2_ acyl chains is now addressed in the legend to Figure 4 and Figure 4—figure supplement 2.